# Operationalizing Leadership and Clinician Buy-In to Implement Evidence-Based Tobacco Treatment Programs in Routine Oncology Care: A Mixed-Method Study of the U.S. Cancer Center Cessation Initiative

Sarah D. Hohl [1,2,*], Jennifer E. Bird [1], Claire V. T. Nguyen [1], Heather D'Angelo [1], Mara Minion [1], Danielle Pauk [1], Robert T. Adsit [1,3], Michael Fiore [1,3], Margaret B. Nolan [1,3] and Betsy Rolland [1,4]

1   Carbone Cancer Center, School of Medicine and Public Health, University of Wisconsin-Madison, Madison, WI 53726, USA; jebird@wisc.edu (J.E.B.); vtnguyen22@wisc.edu (C.V.T.N.); dangeloh@nih.gov (H.D.); minion@wisc.edu (M.M.); danielle.pauk@wisc.edu (D.P.); ra1@ctri.wisc.edu (R.T.A.); mcf@ctri.wisc.edu (M.F.); mbnolan2@ctri.wisc.edu (M.B.N.); brolland@wisc.edu (B.R.)
2   Department of Family Medicine and Community Health, School of Medicine and Public Health, University of Wisconsin-Madison, Madison, WI 53715, USA
3   Center for Tobacco Research and Intervention, School of Medicine and Public Health, University of Wisconsin-Madison, Madison, WI 53726, USA
4   Institute for Clinical and Translational Research, School of Medicine and Public Health, University of Wisconsin-Madison, Madison, WI 53726, USA
*   Correspondence: sarah.hohl@fammed.wisc.edu

**Abstract:** Background: Delivering evidence-based tobacco dependence treatment in oncology settings improves smoking abstinence and cancer outcomes. Leadership engagement/buy-in is critical for implementation success, but few studies have defined buy-in or described how to secure buy-in for tobacco treatment programs (TTPs) in cancer care. This study examines buy-in during the establishment of tobacco treatment programs at National Cancer Institute (NCI)-designated cancer centers. Methods: We utilized a sequential, explanatory mixed-methods approach to analyze quantitative data and qualitative interviews with program leads in the U.S.-based NCI Moonshot-supported Cancer Center Cessation Initiative (*n* = 20 Centers). We calculated descriptive statistics and applied structural coding and content analysis to qualitative data. Results: At least 75% of participating centers secured health care system administrative, clinical, and IT leadership buy-in and support. Six themes emerged from interviews: engaging leadership, access to resources, leveraging federal funding support to build leadership interest, designating champions, identifying training needs, and ensuring staff roles and IT systems support workflows. Conclusions: Buy-in among staff and clinicians is defined by the belief that the TTP is necessary, valuable, and evidence based. Recognizing and securing these dimensions of buy-in can facilitate implementation success, leading to improved cancer outcomes.

**Keywords:** evidence-based tobacco dependence treatment; organizational support and buy-in; implementation science; cancer treatment; cancer survivorship; oncology care; health systems; mixed methods; clinical champion; evidence-to-practice

## 1. Introduction

Approximately 24% of patients with cancer smoke cigarettes at the time of diagnosis [1]. Delivering evidence-based tobacco dependence treatment—i.e., counseling and pharmacotherapy—to those patients improves rates of tobacco abstinence, response to cancer treatment, survival outcomes, and other quality-of-life and cancer-related outcomes [2–4]. Despite this evidence, tobacco treatment programs (TTPs) are inconsistently

implemented in the context of oncology care [5]. In oncology settings, only 50–60% of patients who smoke are advised to quit [1,6], and fewer are referred to treatment [6].

Multiple factors at the patient, clinician, and health system levels influence the success of tobacco dependence treatment program implementation. Patients diagnosed with cancer may not be ready to quit, want to quit smoking on their own [7], or be aware of the benefits of smoking cessation on their cancer outcomes [8]. Clinician-level factors that contribute to suboptimal implementation include perceptions that patients are not interested in quitting, limited time [9], assessing tobacco dependence treatment delivery as a low priority [10], and a lack of skills to deliver it [6]. At the system level, tobacco dependence treatment is often not reimbursable by health insurance in the U.S., and resources and training for treatment delivery are not built into programs [11].

Mitigating such barriers requires obtaining buy-in to successfully implement and integrate TTPs as part of routine cancer care. Buy-in from clinical leadership and oncology clinicians specifically is a critical factor for the implementation success of evidence-based programs in behavioral health and primary care settings [12,13]. Evidence also supports the importance of identifying and training champions—individuals who support and promote implementation by helping others overcome indifference or resistance—as a strategy to obtain and sustain that buy-in [13–16]. Some prior work examining implementation challenges in cancer survivorship care has described buy-in as access to tangible resources, such as financial and physical support [15]. In other implementation studies, buy-in is characterized through support, advocacy, and commitment [13]. However, few prior studies have defined buy-in, and none have examined buy-in in the context of tobacco dependence treatment implementation in cancer care settings. As evidence-based TTPs are scaled-up across cancer care settings globally, this information will guide programs in achieving necessary buy-in and support for implementation and maintenance.

In 2017, the U.S.-based National Cancer Institute (NCI) initiated the Cancer Center Cessation Initiative (C3I) as part of the Moonshot program with the goal of integrating and enhancing TTPs in routine oncology care [17]. Since then, a total of 52 NCI-designated cancer centers have been funded in three successive cohorts. A coordinating center based at the University of Wisconsin-Madison provides technical assistance, evaluation, integration, and knowledge sharing in the initiative [18]. C3I has bolstered the implementation of evidence-based TTPs across a wide and diverse network of cancer care settings, resulting in increased tobacco screening and referrals to treatment, as well as gains in reaching patient groups who have been historically marginalized in the U.S. [19,20]. TTP implementation at each funded cancer center was led by a tobacco cessation researcher, a cancer center clinician, or a combination of both. Participating cancer centers implemented varied TTPs that included diverse patient populations, program models (e.g., tobacco treatment provided by clinicians at the point of care versus referral to an internal or external treatment program), and health system priorities. This provides an ideal opportunity to explore the processes of C3I program leads used to obtain and operationalize buy-in as part of TTP implementation.

## 2. Materials and Methods

### 2.1. Study Design

In this mixed-methods sequential explanatory study [21], we conducted a secondary analysis of quantitative and qualitative data collected for the period July–December 2019 as part of existing C3I evaluation activities. The analysis utilized: (1) quantitative survey data to identify levels of leadership and staff buy-in for evidence-based TTPs across 20 NCI-designated cancer centers from the first C3I cohort; (2) interview transcripts among program leads to explore how they operationalized buy-in. Specifically, we sought to define buy-in and identify its core components and to determine how program leads obtained buy-in and what they perceived as the implications of having buy-in. The use of both quantitative and qualitative data in this sequential explanatory study was designed to assess the ways in which interview data help explain survey data regarding leadership and clinician buy-in

for TTPs across the initiative. This work is reported in alignment with the Consolidated Criteria for Reporting Qualitative Research (COREQ).

*2.2. Data Collection*

2.2.1. Quantitative Data

Every six months, the C3I Coordinating Center collects data from C3I cancer centers via a web-based Qualtrics survey. Quantitative items assess cancer center and TTP characteristics, staffing, the status of implementation activities, and aggregate data of program reach and effectiveness among patients for the prior 6-month period. The biannual report is described in more detail elsewhere [19].

For the current study, to answer our first research aim (i.e., identify levels of leadership and staff buy-in), we analyzed 5 items evaluating implementation activities to assess cancer center/health system leadership and clinician and staff buy-in with the following response options: not applicable, not started, in progress, completed, and in maintenance (Table 1). In addition, given the evidence supporting the value of champions in successful implementation and potentially clinician buy-in [13–15], we evaluated responses from a single item to assess responses to the question: To what extent does your practice have engaged, ongoing champions? Responses to this question were measured on a scale from [1] no extent to [7] full extent.

**Table 1.** C3I survey items to assess leadership and clinician/staff buy-in and presence of clinical champions.

| **Please Indicate the Status of Each of These Activities during This Reporting Period Using the Following Definitions:** |
| --- |
| **Not started:** Work has not begun on this activity. <br> **In progress:** Currently working/devoting significant personnel/resources towards the activity. <br> **Completed:** Goals of the activity have been met; no longer working towards the activity or devoting significant personnel/resources towards the activity. <br> **Maintenance:** Currently devote some time/resources to maintaining a previously completed activity. <br> 1. Secure health care system administrative leadership (e.g., CFO, CEO) buy-in and support <br> 2. Secure health care system clinical leadership (e.g., CMO, cancer center director) buy-in and support <br> 3. Secure health care system information technology leadership (e.g., CIO) buy-in and support <br> 4. Train clinicians and staff in the new clinical workflow <br> 5. Train clinicians and staff to implement the tobacco treatment program |
| **For each statement, select the number that best indicates the extent to which your practice has or does the following things:** <br> The practice has engaged, ongoing champions. No extent (1) to Full extent (7) |

CFO: Chief Financial Officer; CEO: Chief Executive Officer; CMO: Chief Medical Officer; CIO: Chief Information Officer).

2.2.2. Qualitative Data

To address the second aim of our study (i.e., explore how program leads operationalized buy-in), we performed secondary data analysis on interview transcripts of C3I program leads and co-leads. The C3I coordinating center principal investigator (BR) and project scientist (HD), both PhD-level researchers with training in qualitative approaches, conducted in-person interviews with up to 10 program team members at each center between September 2018 and October 2019. The purpose of the 30–60 min interviews was to obtain in-depth perspectives on the TTP and its implementation across C3I centers. The interview guide (available upon request) included 12 main questions with probes to better understand the process, barriers, and facilitators of implementing an evidence-based TTP in oncology care. Interview participants were informed of the purpose of the interviews, which were audio-recorded and professionally transcribed. For this analysis, we included only transcripts from program leads representing the 20 (of 22) Cohort 1 C3I centers that

participated in the qualitative interviews and submitted quantitative report data for the July to December 2019 reporting period. One Cohort 1 C3I center did not submit quantitative report data for this reporting period, and one did not participate in qualitative interviews, so both of those centers were excluded from the current analysis.

The aim of the qualitative analysis was to understand how program leads described buy-in, how buy-in was obtained, and the ways in which levels of cancer center leadership and clinician buy-in influenced TTP implementation. We utilized a homogenous, purposive sampling approach for this analysis [22], in which we selected all program lead interview transcripts, given the leads' intimate knowledge of program implementation, responsibility for obtaining buy-in across cancer center leadership and clinicians/staff, relative role similarity across programs, and experiences with the greatest potential for the inferential transferability of findings to other oncology settings [23].

### 2.3. Data Analysis

Quantitative data were analyzed using SPSS (Version 27). Descriptive statistics were calculated for program characteristics (items are listed in Table 2). The 6 items measuring buy-in and the presence of a clinical champion were used to categorize centers and conduct cross-case analyses [24] in the qualitative phase.

**Table 2.** Cancer center and tobacco dependence treatment program characteristics.

| Characteristic | Mean (*n* or %) | Range |
|---|---|---|
| Number of all patients served at cancer center | 21,220 | 507–95,149 |
| Number of reported patients who smoke | 1911 | 203–4561 |
| Screening rate | 93% | 49–100% |
| Smoking prevalence | 12% | 4–47% |
| Program reach among patients who smoke | 23% | 4–85% |
| Tobacco use treatment program time in operation | *n* | % |
| <2 years | 10 | 50% |
| ≥2 years or more | 10 | 50% |
| Referral strategies | *n* | % |
| Optional EHR referral | 13 | 65% |
| Clinician-initiated referral | 10 | 50% |
| Automatic EHR referral | 8 | 40% |
| Information given patient initiates | 8 | 40% |
| Evidence-based tobacco use treatment types offered | *n* | % |
| Individually delivered in-person counseling | 17 | 85% |
| Pharmacotherapy | 16 | 80% |
| Health system affiliated telephone-based counseling | 14 | 70% |
| Quitline via eReferral or fax | 14 | 70% |
| Point-of-care counseling | 10 | 50% |
| SmokefreeTXT referral | 9 | 45% |
| Group delivered in-person counseling | 5 | 25% |
| Web resource (e.g., Smokefree.gov) | 5 | 25% |
| TelASK or other IVR | 3 | 15% |
| Eligible patients | *n* | % |
| Outpatients | 19 | 95% |
| Inpatients | 7 | 35% |
| Family members | 4 | 20% |
| Engaged, ongoing champions integrated into program | *n* | % |
| Fully | 5 | 25% |
| Somewhat | 14 | 70% |
| Not at all | 1 | 5% |

Transcripts (*n* = 30) of program lead/co-lead interviews across 20 C3I centers that submitted data reports for the June–December 2019 reporting period were uploaded into NVivo (2020) [25]. Two primary coders (JB and CN) performed structural coding, in which a content-based phrase (e.g., leadership buy-in, clinician training, and champions) was applied to segments of text [26]. Coders were trained by and worked under the direction of the lead author (SH), who has expertise in mixed-method research approaches. Researchers double coded 12 (40%) of the interviews until an agreement was reached; any discrepancies were discussed and resolved by consensus during weekly team meetings. As part of a conventional content analytic approach [27], primary coders further reviewed all quotes associated with the buy-in-related codes and developed memos to document specifics of obtaining buy-in and similarities and differences across cases. The research team agreed upon emergent themes and selected representative quotes for inclusion in the manuscript that best illustrate each theme and equitably represent perspectives from all centers.

### 3. Results

*3.1. Cancer Center and Tobacco Treatment Program Characteristics*

Cancer center and TTP program characteristics are reported in Table 2. Between July and December 2019, the mean smoking prevalence across centers was 12%. Across centers, an average of 23% of patients who smoke received evidence-based tobacco treatment. Most centers reported using an optional Electronic Health Record (EHR) referral system (65% of centers) and offering individually delivered in-person counseling (85%). Outpatients were the most common patient group eligible for a TTP across C3I centers. Only five centers (25%) reported having fully integrated TTP champions, with an additional 14 (70%) reporting champions as somewhat integrated. We organized emerging themes from qualitative data into two categories: (1) operationalizing leadership buy-in for initiating and sustaining program implementation and (2) operationalizing clinician and staff buy-in to refer patients and deliver treatment. Table 3 lists each theme, the component of buy-in it represents, and the implications of securing that component for TTP implementation.

**Table 3.** Defining buy-in and its consequences for tobacco dependence treatment programs in oncology care settings.

| EMERGENT THEMES: HOW PROGRAM LEADS OPERATIONALIZED BUY-IN FOR TTPS IN CANCER CARE | CRITICAL COMPONENTS OF BUY-IN | RESULTS OF OBTAINING BUY-IN COMPONENTS |
|---|---|---|
| **Operationalizing leadership buy-in for initiating and sustaining program implementation** | | |
| - Engaging leaders with decision-making power regarding the TTP through meetings and regular communication about the TTP, its needs, and its progress. Engagement included clinical, administrative, IT leadership, and other staff;<br>- Requesting access to leaders' social and financial capital to further support the TTP;<br>- Leveraging support of influential (funding) agencies for the program when communicating with leadership. | Verbal support for the program | - Access to resources and power that helped mitigate implementation challenges (e.g., changes to the EHR and Health IT systems, space to deliver counseling);<br>- Increased program visibility and enhanced integration of tobacco use treatment into routine oncology care. |
| | Communicating value of program, leveraging connections with other leaders within and outside of the cancer center | |
| | Provision of financial resources for:<br>- Clinician and staff FTE;<br>- Clinician and staff education and training;<br>- Evidence-based treatment not covered by health insurance. | - Adequate clinician and staff FTE to effectively implement and sustain the TTP;<br>- Time and financial resources for clinicians and staff to attend training;<br>- Access to networks of other cancer centers and clinicians offering tobacco use treatment in the context of cancer care;<br>- Evidence-based treatment available to patients whose health insurance did not cover treatment. |
| | Commitment of office space | - Designated private spaces to delivery evidence-based counseling to patients with cancer who smoke. |
| | Investment in IT and EHR systems changes and staff time to support new workflows and monitor TTP progress | - Well-functioning, integrated EHR and IT systems to support the TTP; ease of use for clinicians;<br>- IT team able to prioritize workflow integration and mitigate challenges collaboratively with TTP team members. |

**Table 3.** *Cont.*

| EMERGENT THEMES: HOW PROGRAM LEADS OPERATIONALIZED BUY-IN FOR TTPS IN CANCER CARE | CRITICAL COMPONENTS OF BUY-IN | RESULTS OF OBTAINING BUY-IN COMPONENTS |
|---|---|---|
| **Operationalizing clinician and staff buy-in to refer patients and deliver treatment** | | |
| - Designating a program champion to support training and implementation, help address implementation challenges, and build overall support for program implementation; <br> - Identifying training needs and offering, providing access through other institutions, and/or or requiring training on evidence of: | Belief that TTP is necessary, valuable, and evidence-based | - Heightened understanding among leadership, clinicians, and staff of the value of tobacco use treatment as integral to improving cancer health outcomes; <br> - Increased number of patients who are referred and receive evidence-based tobacco use treatment, resulting in improved morbidity and mortality outcomes among patients who smoke. <br> - Clinicians following, rather than ignoring EHR prompts to document patients' smoking status and refer patients to TTP; <br> - Increased referrals of patients with cancer to TTP;Increased implementation of the TTP. |
| ○ TTP effectiveness for quitting smoking among patients with cancer; <br> ○ TTP effectiveness for improving cancer treatment and survivorship outcomes; <br> ○ Utilization of evidence based TTPs among patients with cancer; | Belief that patients served at the cancer center will utilize the TTP | |
| - Offering, providing access through other institutions, and/or or requiring training on how to refer patients and implement the evidence-based TTP; <br> - Leveraging leadership support and IT staff time to ensure referral process and TTP are integrated into existing workflows. | Self-efficacy and willingness to refer patients to the TTP | |
| | Self-efficacy and willingness deliver the TTP to patients with cancer | |

C3I: Cancer Center Cessation Initiative; TTP: Tobacco Treatment Program; FTE: Full-Time Equivalent; IT: Information Technology; EHR: Electronic Health Record.

*3.2. Operationalizing Leadership Buy-In for Initiating and Sustaining Program Implementation*

Figure 1 illustrates the level of buy-in and related activities reported across the 20 centers. At least 75% (≥15 centers) reported securing health care system administrative, clinical, and IT leadership buy-in and support as completed or in maintenance (Figure 1). Three themes emerged regarding operationalizing leadership buy-in as respondents described both the process and outcomes of securing—or not securing—it: (1) engaging leadership; (2) requesting access to resources; (3) leveraging NCI support to build leadership interest in TTPs.

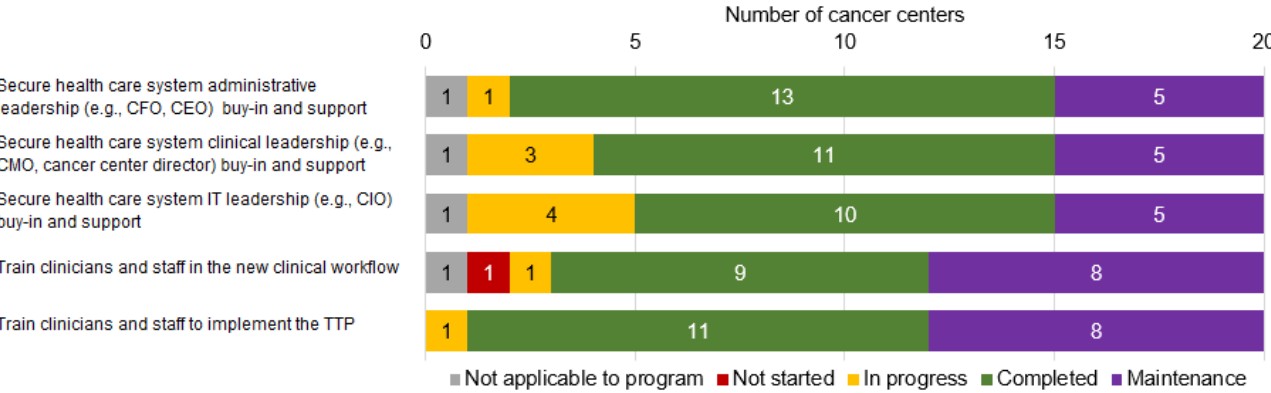

**Figure 1.** Status of implementation activities for tobacco dependence treatment program buy-in at 20 NCI-designated cancer centers, from July to December 2019. CFO: Chief Financial Officer; CEO: Chief Executive Officer; CMO: Chief Medical Officer; CIO: Chief Information Officer).

3.2.1. Engaging Leadership

When describing how they initiated their TTP implementation, respondents from 19 (95%) of the centers highlighted the importance of engaging cancer center, hospital, and clinic leadership early and often throughout implementation. This involved meeting regularly with leadership, describing the value of the TTP for patients with cancer, and requesting access to resources and power to support the program. Most program leads perceived verbal support from cancer center leadership as the foundational step to securing a cascade of buy-in across levels in the health system among critical partners who would help ensure the program's success. This program lead asserted:

> *"The buy-in from the cancer center director was, I think, a critical step in getting administration onboard. And then once administration was onboard, the [human resources] staff were also willing to work with us. Although the time pressures and demands on the EHR staff on the numerable other things that they need to be collecting, I think, puts a little bit of a pressure on this".*

> -Respondent 10, Center 8

Another program leader echoed that gaining this buy-in from cancer center leadership helped expedite this process of further engaging other partners across the health system:

> *"As we were able to get buy-in from the cancer center leadership, the amount of support seemed to move, and the support from the administrative people and from both [hospital] administration and our own departmental administration in being able to do simple things as find the space around the clinic and work out billing issues, all of that moved much faster once we had buy-in from the cancer center leadership, knowing that this was something that was important to the cancer center".*

> -Respondent 15, Center 11

The four centers for whom securing leadership buy-in was not completed described specific challenges regarding turnover throughout the health system that impacted the level of buy-in and the effort required to obtain it, as illustrated by the following program lead:

*"One of the things that hurt us a lot here is that during the past two years, we have gone through a lot of changes and transition in leadership. So, every time we have a meeting with leadership members, by the time we go back, the [previous leadership is] not there or they have forgotten [ … ] Every time, it was like starting from scratch, and a new approval is required from someone that we never heard of before".*

-Respondent 6, Center 5

### 3.2.2. Requesting Access to Resources

*"We need leadership to provide resources. We need institutional support for these two full-time positions and to cover nicotine replacement therapy and to cover the space".*

-Respondent 1, Center 1

Beyond obtaining verbal support for the TTP, program leads across all 20 centers said leadership buy-in was demonstrated through the allocation of tangible resources, such as staff time—including IT staff time, which is often a shared resource at cancer centers—designated office space within the cancer clinics, and funds for evidence-based treatment that was not reimbursed by health insurance. Program leads saw their own role as asking for what they needed from clinic and cancer center leadership to implement and sustain a successful TTP, even when those asks required significant time and effort. For example, this program lead explained that the

*" … biggest test of this program going forward is, Do we have enough staff? We'll find out very quickly. [ … ] For now, we'll get the main leadership buy-in to go to their faculty meetings, go to their nurse meetings. That's basically going to be a full-time job for all of probably January and February".*

-Respondent 28, Center 9

Program leads said that integrating referral systems and the TTP into routine workflows and monitoring its progress required significant IT support. More than a third of program leads indicated that their IT teams prioritized time-sensitive requests from other cancer center programs over those of the TTP, creating delays and impeding progress towards implementation. Program leads said that identifying IT leaders who would advocate for the program was an essential step in obtaining buy-in from those leaders and their teams. A program lead at one center who had engaged the IT team successfully described the cancer center IT director as being:

*" … like our quarterback in the IT world. [ … ] He just knows who to talk to. They had their annual system-wide IT meeting [IT Director] invited us to present about this program because he noted IT people usually work behind the scenes. They don't always learn how their work impacts patient care. After the presentation, his words literally were, 'My inbox is flooded with people wanting to know how they can get involved in this project'".*

-Respondent 8, Center 6

In 2019, 17 (85%) centers were offering in-person evidence-based counseling, which required designated physical space to deliver that treatment. At most cancer centers, the TTP spanned multiple clinics where space was a limited resource. However, the relationships and buy-in program leads had established with clinic and cancer center leadership helped mitigate space issues, as illustrated by the following program lead:

*"The director of the ambulatory operations in our cancer clinics controls the space in our hematology-oncology clinics, which is where they have like a rotating schedule of space that [the tobacco treatment specialist] can use. If it becomes a problem, she will reach out very practically and let him know like, 'Hey, this is what happened today'. [ … ] They're supporting us in finding that space. [ … ] We have her set up to have a mobile office capability [with a] laptop. [Cancer center director] got her a cellphone for use inhouse".*

-Respondent 7, Center 6

### 3.2.3. Leveraging NCI Support to Build Leadership Interest in TTPs

A total of 17 program leads representing 15 (75%) centers highlighted the critical role that funding and program endorsement from NCI played in bolstering cancer center leadership support and facilitating early implementation success. Of those, over half described specifically leveraging NCI's backing of the program as a key discussion point in meetings with cancer center leadership, who program leads perceived to be highly motivated to demonstrate progress in NCI priority areas. This respondent explained that achieving implementation success for the TTP was now an institutional priority:

*" ... I've been doing this for a really long time and have had a very hard time convincing people that tobacco dependence services are a critical part of a health system's role. As soon as the NCI spoke up, that changed".*

-Respondent 21, Center 15

Program leads described NCI funding itself as a net benefit to initiating the implementation of TTPs at their centers, particularly to designate staff time to develop and scale-up successful programs while devising a longer-term strategy for sustainability. Illustrating this concept, this program lead offered:

*"The NCI being willing to be involved in this and to put in, even if only a small amount of money to get this jump started, is a very big deal. [ ... ] Without that, we wouldn't have seen the program success that has happened in this past year. Now I'm fully confident that we can continue to keep this up and running over the next years to decades".*

-Center 8, Respondent 10

However, the end of the grant period presented difficulties at some centers, regardless of if they reported having secured leadership buy-in, as staff would need to be funded outside of the grant model on which the program originated. This program lead explained:

*"We had a long, arduous negotiation with the hospital over making good on their commitment in the proposal to hire a [tobacco treatment specialist]. [ ... ] It required the addition of an FTE to their hard budget ... to absorb the people who had already been hired and had been paid for four years on grant funds. It's just, you know, anything they don't have to, if they can't get reimbursement and it's going to cost them money, they're skeptical".*

-Respondent 3, Center 2

Yet, the persistent engagement of cancer center leadership enabled program leads to begin conversations about securing funding beyond that provided by NCI. This program lead explained:

*"We knew we would have to have [cancer center administrative and clinic directors] on board. [ ... ] It's really not hard to sell treating tobacco [dependence] to oncologists, you know, at least in the head-nod version of doing it. Now, when it comes to the wallet version, I'm not so sure".*

-Respondent 16, Center 12

### 3.3. Operationalizing Clinician and Staff Buy-In to Refer Patients and Deliver Treatment

Between 85 and 95% of centers had trained clinicians and staff in the new clinical workflow and to implement the TTP (Figure 1). Three themes emerged regarding operationalizing clinician and staff buy-in: (1) designating program champions; (2) identifying education and training needs; (3) ensuring IT systems support TTP workflows.

### 3.3.1. Designating Program Champions

In quantitative surveys, 19 (95%) centers reported having fully or somewhat integrated clinical champions into the TTP; however, respondents at just 12 (60%) centers discussed champions during qualitative interviews. Program leads representing those centers emphasized champions' expertise in building enthusiasm among their peers for the program,

which they accomplished through leading training and education, answering questions, and helping clinicians overcome TTP ambiguity and indifference. A few program leads also asserted that champions played a pivotal role in achieving buy-in from cancer center and clinic-level leadership in addition to that from clinicians and staff. Program leads said champions were designated this role because they were trusted and respected among their peers and leadership, knowledgeable about tobacco dependence treatment, and dedicated to its implementation. They described two ways that champions were identified. At some centers, program leads recognized promise in nurse practitioners or oncologists themselves. For example, this program lead explained:

> *"The cancer clinicians are so consumed by the active treatment, they weren't considering tobacco as part of treatment. This project is a tremendous opportunity to change that culture [ . . . ] When this project came up, I said, '[champion name], you seem to get it, and you seem to be a perfect person for this sort of project.' [ . . . ] There was no question she is someone who was able to have relationships and come as an insider and bring this part in to start changing the culture".*

-Respondent 23, Center 16

In contrast, at other centers, program leads relied on the relationships they had developed with cancer center leadership to recommend a person who fit the criteria stated above. This respondent described that the cancer center director:

> *" . . . was really instrumental in identifying a champion for the program. [We] said, 'You know, who do you recommend?' And he recommended two of the very best champions in the cancer center, one on the medical side, one on the radiation side, who have been part of this from the beginning".*

-Respondent 2, Center 2

### 3.3.2. Identifying Education and Training Needs

Regardless of the presence of champions at their centers, program leads emphasized the importance of providing clinicians and staff—and when possible, cancer center leadership—with opportunities for training to encourage buy-in and increase the likelihood, frequency, and quality of TTP implementation. Program leads highlighted four critical areas in which they perceived that clinicians and staff required ongoing education and training. First, they perceived a need for education on the evidence for tobacco dependence treatment effectiveness among patients with cancer. For example, this program lead recalled:

> *"Some of the people that have been here for a very long time have kind of hardened to the idea that smoking is something that they're going to be able to change in our cancer patient population and in our culture. They're pessimistic, and it's just not a priority".*

-Respondent 15, Center 11

Second, program leads observed that leadership, clinicians, and staff were largely unaware of evidence for the effectiveness of tobacco cessation in improving cancer treatment and survivorship outcomes. This program lead articulated:

> *"One of the other challenges is just getting people to understand the importance of intervening early in a person's cancer experience. [ . . . ] They're still worrying about, 'Let's get to survivorship, and then we'll start talking about this,' which the evidence doesn't support that. I mean, they're less likely to get to survivorship, is what we've been pointing out, if we wait until then".*

-Respondent 1, Center 1

Third, program leads remarked that many providers were skeptical that patients with cancer would participate in tobacco dependence treatment, given these patients' unique considerations, such as treatment side effects and smoking as means for coping with a cancer diagnosis. Finally, program leads said clinicians and staff required training on how

to refer patients and effectively implement the TTP. Training sessions to address these four areas were implemented in a multitude of ways across cancer centers. At one center, for example, program leads utilized weekly in-services among clinicians

" ... *to increase awareness of the program and create a more personalized relationship with the oncologists to be able to get those referrals".*

-Respondent 6, Center 5

Champions also commonly delivered training, relying on the trust and relationships they had established among peers to garner interest in program implementation. Several program leads described leveraging the C3I network to send staff to training offered at other C3I centers or to develop joint training programs. For example, this program lead asserted:

*"We were very fortunate in that we paired up with [other C3I center] to create the tobacco treatment specialist program, and now all of our providers go through that program. We provide them specialized training, which we've had to develop for medical providers, specialized training for behavioral providers, and then a lot of just hand holding as well".*

-Respondent 9, Center 7

### 3.3.3. Ensuring Staff Roles and IT Systems Support TTP Workflows

Program leads were cognizant that clinician and staff buy-in for the TTP was influenced by who was responsible for referring patients, how easy the referral process was to use, and how well the TTP was integrated into their clinical workflows. In 2019, fewer than half of centers were utilizing automatic EHR referrals; most centers utilized either optional EHR or clinician-initiated referrals (Table 1), which required clinicians or staff to both screen patients for smoking and manually refer those who smoke to the TTP. Many program leads recognized—or were told by their cancer center leadership—that oncologists were unlikely to screen and refer patients. For example, this program lead offered:

*"I think for us, the biggest problem is the rate at which the oncologists are canceling the [referral]. I think it's upwards of two-thirds, so we've still got a long way to go. And many of them may be new patients or inpatients".*

-Respondent 21, Center 15

Many program leads said they mitigated this challenge by giving the responsibility for referrals to nurses, medical assistants, or tobacco treatment specialists. Illustrating this theme, this program lead described hiring a full-time nurse practitioner who

" ... *would show the physicians the best practice advisory and orient them to how to prescribe. However, we'd say, 'If you don't want to prescribe, we're happy to, but let's set up [electronic] links so that we can see these patients'".*

-Respondent 30, Center 20

Program leads observed that unless the referral process was both perceived as important and easy to use, clinicians would likely not use it. As this program lead reflected:

*"We don't have an alert culture. Physicians ignore it anyway. I will admit to that also. You know how to bypass it, and you'll just ignore it".*

-Respondent 23, Center 16

Program leads who perceived themselves as the most successful in addressing clinician resistance persistently engaged their cancer centers' IT leadership and staff and facilitated exchange between IT teams and the clinicians. This program lead explained:

*"I was surprised how willing and how supportive [the IT staff] were in the process of getting providers engaged in the tobacco treatment program. [They pulled] some of the data on the Quality Oncology Practice Initiative, analyzed the workflows. They even made small tests of change to support implementation".*

-Respondent 14, Center 10

## 4. Discussion

In this sequential, explanatory mixed-methods study, we utilized data from quantitative reports and qualitative interviews with 30 program leads at 20 cancer centers throughout the U.S. participating in the Cancer Center Cessation Initiative. Centers offered a range of evidence-based tobacco dependence treatment interventions in inpatient and outpatient clinics, including counseling, pharmacotherapy, and referrals to Quitlines. In quantitative reports, 75% or more centers reported having secured multiple levels of leadership buy-in and having trained clinicians/staff in areas relevant to the TTP. Qualitative data further elucidated components of buy-in across levels of cancer center leadership and clinician/staff levels that were critical to ensuring the successful implementation of TTPs in cancer care settings and the implications of securing each component for TTP implementation (Table 3). These components both help define what it means to obtain buy-in and offer tangible indicators for program leads in other oncology settings to monitor progress and prioritize efforts as they initiate or enhance TTP implementation.

Our work both corroborates and builds on prior studies that have examined leadership and clinician buy-in for implementing new programs in primary care, behavioral health, and oncology settings. Published work in implementation science has characterized buy-in as providing support and advocacy [13]. In a study examining cancer survivorship program implementation in primary care, buy-in has been characterized as access to resources such as staff time and health IT support [15]. Our work adds dimensions to this conceptualization of buy-in and describes how program leads acquired it. At the leadership level, the minimum indication of buy-in was verbal support for the TTP, followed by leadership communicating the TTP's value throughout the cancer center and leveraging their connections to secure power and resources to support it. Program leads obtained buy-in by identifying which leaders at their cancer centers had power to provide resources that met the TTP's needs, leveraging NCI's prioritization of the program and its success in regular communication with leadership, and requesting access to the resources they needed. Leadership and staff turnover emerged as a roadblock to obtaining buy-in and required program leads at those centers to spend significantly more time engaging, training, and communicating with new staff about the TTP.

Prior work suggests that implementing new programs for cancer patients and survivors requires buy-in characterized by the belief that this population needs the program and would use it [15]. Moreover, overcoming clinician resistance to new programs is critical to their implementation and success [9–11]. In our work, leadership and clinician/staff buy-in was further characterized by these groups' beliefs that the TTP was necessary, valuable, and evidence based, and that patients with cancer would utilize the program. Over half of the program leads indicated that designating clinical champions helped educate clinicians—and occasionally leadership—about the evidence surrounding these concepts. Clinical champions have been vital to the successful implementation of evidence-based programs within behavioral health and cancer care settings, as they are often charged with leading training to improve clinicians' knowledge on the need for programs and self-efficacy to implement them [28–30]. In addition to designating champions, program leads leveraged the network, resources, and expertise of other NCI-designated cancer centers participating in C3I to meet staff/clinician training needs, a unique aspect of the C3I program.

Program leads described the process of obtaining buy-in from multiple levels as iterative in nature. Although it was possible to have leadership buy-in without staff/clinician buy-in and vice versa, program leads who had secured cancer center leadership's verbal support for the program, received financial resources and office space, and were provided IT support to implement new workflows and program monitoring indicated that securing staff buy-in came more easily. Centers that did not report having secured all three levels of leadership buy-in tended to focus more on obtaining buy-in among staff and clinicians who would implement the program, particularly as some managed leadership turnover. Future research is needed to determine the extent to which buy-in at the leadership and clini-

cian/staff level influenced the longer-term program sustainability, reach, and effectiveness of programs in C3I.

Our study is the first to examine buy-in across a diverse network of cancer centers committed to integrating tobacco dependence treatment into routine oncology care. We used both quantitative and qualitative data to characterize and operationalize this concept. Despite its strengths, there are some limitations to this work. No survey item specifically assessed clinician buy-in, nor did our analysis differentiate between clinician and staff buy-in. The two questions regarding training served as proxies for staff and clinician buy-in. This is consistent with prior work that has found that training improves buy-in among clinicians and those implementing or supporting a new program [12,31]. The 20 NCI-designated cancer centers in this analysis represent those with specific funding to integrate and enhance TTPs among patients and were part of a national network of cancer centers, supported by a central coordinating center to do so. Thus, they may not be representative of all cancer care settings. However, the components of buy-in are likely transferrable to multiple diverse oncology care settings.

## 5. Conclusions

Obtaining buy-in from cancer center and clinical leadership is critical to the successful implementation of tobacco dependence treatment into routine oncology care, but buy-in has not been clearly defined or operationalized. Despite 75% or more of the centers reporting having secured all levels of buy-in and training in quantitative reports, nuances emerged in qualitative data, which offered a deeper understanding of the what, how, and why of buy-in. Our findings suggest that buy-in from leadership is defined as verbal support, access to tangible resources (e.g., financial, space), and access to power (e.g., leveraging connections to secure resources) that facilitate TTP implementation. Buy-in among staff and clinicians is defined by the belief that the TTP is necessary, valuable, and evidence based; the belief that patients served will utilize the TTP; and self-efficacy and willingness to refer and deliver evidence-based tobacco dependence treatment to patients with cancer who smoke. Securing these dimensions of buy-in are likely to facilitate implementation success, leading to increased rates of cessation and improved short- and long-term tobacco and cancer outcomes for patients.

**Author Contributions:** Conceptualization, Methodology, and Supervision: S.D.H.; Formal Analysis, S.D.H., J.E.B. and C.V.T.N.; Data Collection: B.R. and H.D.; Writing—Original Draft Preparation, S.D.H., J.E.B. and C.V.T.N.; Writing—Review and Editing, M.M., D.P., H.D., M.F., R.T.A., M.B.N. and B.R.; Project Administration, M.M. and D.P.; Funding Acquisition, B.R. All authors have read and agreed to the published version of the manuscript.

**Funding:** This work was funded by the National Cancer Institute (ICF Contract #17GZSK0031).

**Institutional Review Board Statement:** Ethical review and approval were waived for this study due to this work being categorized as routine program evaluation and IRB-exempt.

**Informed Consent Statement:** Participant consent was waived as this program evaluation involved no procedures for which written consent is normally required outside of a research context.

**Data Availability Statement:** The data collected as part of the Cancer Center Cessation Initiative are not publicly available.

**Acknowledgments:** The authors would like to thank program leads at the following universities and affiliated NCI-designated cancer centers who completed C3I reports and provided their time and insight as interview participants: Baylor University, Case Western Reserve University, Duke University, Georgetown University, Indiana University, New York University, University of North Carolina-Chapel Hill, University of California-Davis, University of Chicago, University of Colorado-Denver, University of Kansas, University of Kentucky, University of Minnesota, University of New Mexico, University of Pennsylvania, University of Utah, University of Virginia, Vanderbilt University, Washington University, and Yale University.

**Conflicts of Interest:** The authors declare no conflict of interest.

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
