# Peer review of "Operationalizing Leadership and Clinician Buy-In to Implement Evidence-Based Tobacco Treatment Programs in Routine Oncology Care: A Mixed-Method Study of the U.S. Cancer Center Cessation Initiative"

_curroncol, doi:10.3390/curroncol29040195_

Round 1

Reviewer 1 Report

This is a well written and important study in the first 20 institutions in the NIH supported program. I have the following comments and suggestions

In the results, what was the level or administrative or clinical leaders who gave buy in or support, and which levels were most effective? Was it highest (president, ceo, board level) or upper-mid level (VP, center or division directors, or department chairmen).

In results or discussion was securing buy in or support from physicians/NP individuals different from securing buy in or support from nurses or clinic administrators

In results, did reach of cancer patients who were smokers vary by auto EHR referral vs other referral? by length of time TTP had been in operation? by by degree of administrative or clinical leadership buy in? by degree of training of clinicians?

In discussion, these 20 institutions were grant supported. How applicable do the authors think these results are to non-grant supported institutions where resources may be less or scarce?

What priorities do the authors suggest for implementing tobacco control programs in institutions just starting such programs?

Author Response

We thank the reviewer for their comments, questions, and suggestions. We have done our best to respond to these critiques and made changes in the manuscript, as described in the below.

This is a well written and important study in the first 20 institutions in the NIH supported program. I have the following comments and suggestions.

Thank you. We appreciate your comments and suggestions.

In the results, what was the level or administrative or clinical leaders who gave buy in or support, and which levels were most effective? Was it highest (president, ceo, board level) or upper-mid level (VP, center or division directors, or department chairmen).

Thank you for this question. We have provided examples of positions of individuals in each of the leadership roles from whom buy-in was sought in Table 1 and Figure 1.

In results or discussion was securing buy in or support from physicians/NP individuals different from securing buy in or support from nurses or clinic administrators

This is an excellent question, and one we did not explicitly evaluate in our analysis. We have added a sentence to the limitations section to address this.

In results, did reach of cancer patients who were smokers vary by auto EHR referral vs other referral? by length of time TTP had been in operation? by by degree of administrative or clinical leadership buy in? by degree of training of clinicians?

Another great question. We did not include reach and effectiveness data in this analysis. However, some of our other work has examined the first two (EHR referral, length of time). (See D’Angelo et al 2022: https://doi.org/10.1093/tbm/ibac009) In future work we hope to evaluate the association between implementation outcomes (e.g., reach, effectiveness, sustainability) and buy-in at multiple levels.

In discussion, these 20 institutions were grant supported. How applicable do the authors think these results are to non-grant supported institutions where resources may be less or scarce?

We address this in the last paragraph of the discussion before the conclusion, stating “The 20 NCI-designated cancer centers in this analysis represent those with specific fund-ing to integrate and enhance TTPs among patients and were part of a national network of cancer centers, supported by a central coordinating center to do so. Thus, they may not be representative of all cancer care settings. However, the components of buy-in are likely transferrable to multiple diverse oncology care settings.”

What priorities do the authors suggest for implementing tobacco control programs in institutions just starting such programs?

The emergent themes in Table 3 are meant to serve as guidance for centers who want to begin building their own TTPs.  (e.g., engage leaders, request access to capital, leverage support, etc.)

Reviewer 2 Report

Please , the first line and second-line treatment options in smoking cessation should be addressed in the introduction and discussion.

Data concerning comorbidities and the most types of cancer found in the analysis should be included becasuse the type of cancer and symptoms can affect the motivation to quit smoking.are there data available on the interaction between smoking and oncology drugs used?

Was the brief advice mainly used and by what healthcare provider? 

I suggest to include the following references to discuss about the influence of smoking cessation in cancer patients:

-Future Sci OA. 2019 May 3;5(5):FSO394. doi: 10.2144/fsoa-2019-0017.

-Future Oncol. 2016 Sep;12(18):2149-61. doi: 10.2217/fon-2015-0055

-Comput Methods Programs Biomed. 2022 Apr;216:106660. doi: 10.1016/j.cmpb.2022.106660.

Author Response

We thank the reviewer for their comments, questions, and suggestions. We have done our best to respond to these critiques and made changes in the manuscript, as described in the below in bold text.

Please, the first line and second-line treatment options in smoking cessation should be addressed in the introduction and discussion.

Thank you for this suggestion. We have added examples of evidence-based tobacco treatment in the second sentence of the introduction. These first-line evidence-based tobacco treatment modalities are also listed in Table 2.

Data concerning comorbidities and the most types of cancer found in the analysis should be included becasuse the type of cancer and symptoms can affect the motivation to quit smoking.are there data available on the interaction between smoking and oncology drugs used?

Our analysis does not include any patient-level data, nor did we assess smoking or types of treatment any patients received. This manuscript is about obtaining buy-in to adopt and implement evidence-based tobacco treatment programs in cancer centers.

Was the brief advice mainly used and by what healthcare provider? 

The types of evidence-based treatment offered across all centers are listed in Table 2. Individual counseling was offered across the greatest number of centers. At most centers, tobacco treatment specialists provided that service.

I suggest to include the following references to discuss about the influence of smoking cessation in cancer patients:

-Future Sci OA. 2019 May 3;5(5):FSO394. doi: 10.2144/fsoa-2019-0017.

This manuscript does not discuss smoking cessation. Rather, it discusses the impact of cigarette smoking on tumor spread, which is important, but not relevant to our work.

-Future Oncol. 2016 Sep;12(18):2149-61. doi: 10.2217/fon-2015-0055

We have added a reference to this manuscript in the second paragraph of the introduction.

-Comput Methods Programs Biomed. 2022 Apr;216:106660. doi: 10.1016/j.cmpb.2022.106660.

The above manuscript describes a prediction model for lung cancer risk by smoking history. Since our manuscript is about treatment for patients with all types of cancer who smoke, this manuscript is not directly relevant to our work.